# STAT3 and Its Targeting Inhibitors in Oral Squamous Cell Carcinoma

**DOI:** 10.3390/cells11193131

**Published:** 2022-10-05

**Authors:** Mingjing Jiang, Bo Li

**Affiliations:** 1Jilin Provincial Key Laboratory of Oral Biomedical Engineering, Department of Oral Anatomy and Physiology, Hospital of Stomatology, Jilin University, Changchun 130021, China; 2Liaoning Provincial Key Laboratory of Oral Diseases, Experimental Teaching Center, School and Hospital of Stomatology, China Medical University, Shenyang 110001, China

**Keywords:** STAT3, oral squamous cell carcinoma, oncogene, small molecule inhibitors

## Abstract

Oral squamous cell carcinoma (OSCC) usually originates from the precancerous lesions of oral mucosa and accounts for approximately 90% of oral cancers. Current therapeutic approaches do not yet meet the needs of patients, and the 5-year survival rate of patients with OSCC is only 50%. Recent studies have revealed that the signal transducer and activator of transcription 3 (STAT3) plays a key role in the development and progression of OSCC. STAT3 is overexpressed and constitutively activated in OSCC cells, and promotes cancer cell proliferation and anti-apoptosis, migration and invasion, angiogenesis, radiotherapy resistance, and immune escape, as well as stem cell self-renewal and differentiation by regulating the transcription of its downstream target genes. Inhibitors of the STAT3 signaling pathway have shown the promising anticancer effects in vitro and in vivo, and STAT3 is expected to be a molecular target for the treatment of OSCC. In this review, we highlight the oncogenic significance of STAT3 in OSCC with an emphasis on the therapeutic approaches and effective small molecule inhibitors targeting STAT3. Finally, we also propose the potential research directions in the expectation of developing more specific STAT3 inhibitors for OSCC treatment.

## 1. Background

Oral squamous cell carcinoma (OSCC) represents the most frequent form of head and neck squamous cell carcinoma that is the sixth most common group of cancers worldwide [1,2]. OSCC originates from the tongue, palate, floor of mouth, alveolar ridge, buccal mucosa, and other areas of oral cavity, and accounts for about 90% of oral malignancies [3,4]. In 2020, about 377,713 new patients were diagnosed with lip and oral cavity cancers, and its number of deaths was around 177,757 worldwide. Most cases were discovered in Asia [5]. The risk factors for OSCC include smoking, excessive alcohol consumption, and betel nut chewing, exposure to carcinogens, immunodeficiency, irradiation, nutrition, and genetic susceptibility, as well as viral infections including human papillomavirus and herpes simplex virus [6]. The main biologic activity of OSCC is classified as highly, moderately, or poorly differentiated along with increased aggressiveness [7]. Histologically, OSCC exhibits grades ranging from well-differentiated keratinizing carcinoma to undifferentiated nonkeratinizing carcinoma, which is more apt to spread [8,9]. Patients with OSCC are asymptomatic in the early stages, and most patients are diagnosed when OSCC further progresses, resulting in the lower survival rate [10]. Tumor infiltration, lymph node metastasis, and high rates of local recurrence are the main factors leading to death in patients with OSCC [11]. Current treatment options for OSCC include surgery, chemotherapy, radiotherapy, or a combination of therapies, depending on factors such as the extent of the disease and the patient’s comorbidities [12]. However, the adverse effects of the treatment still exist. For example, salivary gland hypofunction is a common and permanent adverse effect of radiotherapy to the head and neck [13]. The common complications after selective neck dissection are spinal accessory nerve damage and shoulder dysfunction [14]. Systemic administration of chemotherapeutic drugs emphasizes the need to avoid the systemic undesired side effects. Targeted therapy for OSCC, which consists of immunotherapy, gene therapy, and bionic technology, has shown some promise in preliminary clinical studies, but further investigation is needed [15].

The signal transducer and activators of transcription (STAT) family are potential cytoplasmic transcription factors, including STAT1, STAT2, STAT3, STAT4, STAT5a, STAT5b, and STAT6, which can be activated in response to cytokine and growth factor stimulation to mediate multiple intracellular signaling pathways [16]. STAT proteins are characterized by six functionally conserved domains. The N-terminal domain (NTD) mediates the formation of an anti-parallel dimer of un-phosphorylated STAT3; coiled-coil domain (CCD) is used to interact with regulatory proteins that positively or negatively regulate transcriptional activity; DNA-binding domain (DBD) can recognize specific DNA sequences of target genes; The linker domain (LD) is involved in nuclear export and DNA binding; the SH2 domain (SH2) is the most conserved STAT domain, involved in dimer formation, and plays a key role in signal transduction by binding to phosphorylated tyrosine residues of the receptor; the transcriptional activation domain (TAD) is a highly variable domain in length and sequence of STAT3, and regulates the transcriptional activation of target genes by interacting with other transcription factors [17]. The most widely studied one among the STAT family is STAT3, which consists of 770 amino acids and shares similar functional domains with other STAT family members. STAT3 can transmit signals from the cell membrane to the nucleus to activate target gene transcription and regulate a variety of cellular physiological activities, including cell proliferation, differentiation, apoptosis, angiogenesis, and immune system regulation [18]. Although STAT3 plays a crucial role in normal cells, constitutive activation of STAT3 in most human malignancies drives transcription of unscheduled genes and the transcription products subsequently promote tumor progression [19].

Recently, researchers have found that STAT3 is overexpressed and constitutively activated in OSCC and plays an important role in OSCC aggressiveness [20,21,22]. Growing evidence also suggests that STAT3 may be a potential molecular target and biomarker of OSCC, and STAT3 inhibitors have shown efficacy in inhibiting OSCC tumor growth and metastasis [23,24]. Thorough understanding of the roles of STAT3 in OSCC will facilitate the development of STAT3-targeted therapeutics. In this review, we focus on the protumor role of STAT3 in OSCC, and summarize the therapeutic strategies and representative small molecule inhibitors targeting STAT3. Finally, we outline more approaches to target STAT3 founded in other cancers, which may propose potential directions for further research to develop more specific STAT3 inhibitors for OSCC treatment.

## 2. The STAT3 Signaling Pathway

STAT3 is tightly regulated by negative modulators to maintain an inactive state in an unstimulated cell [25]. These negative modulators include members of the protein inhibitor of activated STAT (PIAS), suppressor of cytokine signaling (SOCS) family, cellular phosphatases (SHP1, SHP2, PTPN1, PTPN2, PTPRD, PTPRT, and DUSP22), as well as ubiquitin enzymes. The control of endogenous negative regulators can maintain the STAT3 signal as balanced and stable to perform physiological functions in normal cells [19]. The classical STAT3 signaling pathway is activated by the binding of cytokines or growth factors to their corresponding cell surface receptors. Extracellular ligands bind to cell surface receptors to form dimers, which then recruit and further activate Janus tyrosine kinase (JAK) [26,27]. Cytoplasmic tyrosine residues of the receptor are phosphorylated by activated JAKs to further recruit STAT3 to the phosphorylated tyrosine residues of the cytoplasmic receptor by interacting with the SH2 domain of STAT3. STAT3 is phosphorylated at tyrosine-705. Then, phosphorylated STAT3 (p-STAT3) monomers form homodimers via interactions between phosphorylated tyrosine-705 and the SH2 domain. STAT3 dimers further translocate to the nucleus and bind to DNA enhancer regions in a sequence-specific manner. This induces transcription of target genes critical for physiological and pathological functions [25,28]. Moreover, non-receptor tyrosine kinases such as Src and Abl can also lead to constitutive activation of STAT3 [29].

In the classical STAT3 signaling pathway, the p-STAT3 dimer is able to bind to the corresponding site on DNA and initiates nuclear transcription. Indeed, STAT3 is also localized in mitochondria as a monomer. Mitochondrial STAT3 (MitoSTAT3) can regulate complexes I and II to play a role of modulator for mitochondrial respiration [27,30]. MitoSTAT3 was shown to increase the activities of complex II, ATP synthase (complex V), and lactate dehydrogenase to maintain the glycolytic and oxidative phosphorylation, which contributed to the RAS-dependent oncogenic transformation of mouse embryonic fibroblasts, and inhibition of MitoSTAT3 stopped tumor growth [31,32]. MitoSTAT3’s improved bioenergetics may speed up early neoplastic lesions [33,34]. Unexpected STAT3 localization to the endoplasmic reticulum (ER) has also been discovered. STAT3-mediated IP3R3 downregulation in the ER was a major factor in anti-apoptotic effects of breast cancer cell lines [35]. These results suggest that a further non-canonical function played by constitutively phosphorylated STAT3 at its S727 residue promotes the growth of cancer and inhibits apoptosis, but the mechanism remains to be clarified [31,35,36]. The characteristics of activated STAT3 have been described above; however, it has also been discovered that inactivated STAT3 has bioactive properties. Unphosphorylated STAT3 monomers or dimers recognized specific DNA structures, which was significant for chromatin organization [37]. After IL-6 treatment, STAT3-mediated change of chromatin structure was significantly increased, which played a role in inflammation-mediated cancers [26]. However, the oncogenic role of unphosphorylated STAT3 needs further investigation. 

## 3. The STAT3 Signaling Pathway in OSCC

Studies have shown that overactivation of STAT3 in many types of tumors may depend on the following mechanisms: excessive stimulation caused by overexpressed cytokines and growth factors such as interleukin-6 (IL-6), interleukin-10 (IL-10), epidermal growth factor (EGF) as well as fibroblast growth factor (FGF); hyperactivation of receptors for cytokines or growth factors; elevated activity of cytoplasmic non-receptor tyrosine kinases, such as Src and Abl kinase; loss of negative regulation for STAT3 caused by inactivation or decreased expression of endogenous negative regulators, e.g., PIAS, SOCS, and PTPs. These mechanisms can induce uncontrolled cell growth, malignant cell transformation, angiogenesis, metastasis, invasion, and immune escape [38]. Importantly, STAT3 is overexpressed and constitutively activated in OSCC, which is highly related to OSCC initiation and progression (as shown in Figure 1). For example, a recent study verified that STAT3 was a direct target of mir-125b, and circPVT1 and LncRNA MALAT1 may promote OSCC cell growth by sponging mir-125b and increasing STAT3 expression [39,40]. In addition, endogenous negative regulators such as PTPN4, SOCS5 and SOCS6 were decreased in OSCC cells, which contributed to the upregulation of p-STAT3. The upregulated p-STAT3 promoted the OSCC progression [41,42]. Emerging research demonstrates the critical role of STAT3 in OSCC [20,24,43,44,45,46,47,48,49,50]; we herein present a comprehensive overview of its oncogenic functions in this section.

### 3.1. Role of STAT3 in OSCC Cell Proliferation and Anti-Apoptosis

Several studies have demonstrated that STAT3 promotes cell proliferation and inhibits apoptosis in OSCC by increasing the expression of target genes, including survivin, Mcl-1, c-Myc, Glut5, cyclin D1, B-cell lymphoma-2 (Bcl-2), and B-cell lymphoma extra -large (Bcl-xL) [22,57,58,59]. In OSCC, STAT3 directly bound to the c-Myc promoter and promoted its transcription [60,61]; this can be blocked by silencing upstream musashi RNA-binding protein 1 (MSI1) of STAT3 signaling [62]. LncRNA P4713 promoted OSCC cell proliferation via activating the JAK/STAT3/cyclinD1 pathway. After downregulation the expression level of P471, STAT3 was found to translocate from nucleus to cytoplasm accompanied by a decrease in phosphorylation levels, which attenuated tumor cell proliferation and migration capacity [63]. *Porphyromonas gingivalis* and *Fusobacterium nucleatum* were responsible for the upregulation of cyclin D1 via IL-6/STAT3 dependent-mechanism to drive the OSCC growth [64]. MiR-769-5p restrained the Bcl-2 protein level and increased the protein levels of Bcl-2 associated X protein (Bax) and cleaved-caspase 3 by inhibiting JAK1/STAT3 activity, suggesting the important role of STAT3 in promoting the OSCC growth [65]. Furthermore, STAT3 can promote the transcription of HNF1A-AS1 in OSCC cells, HNF1A-AS1 in turn activated the Notch signaling pathway to promote OSCC cell proliferation. Conversely, depletion of HNF1A-AS1 induced apoptosis and cell cycle arrest [66].

### 3.2. Role of STAT3 in OSCC Cell Migration and Invasion

Epithelial–mesenchymal transition (EMT) is an important biological mechanism underlying the metastasis of primary tumors [67]. During the EMT, the characteristic of epithelial cells will convert from epithelial cells highly expressing epithelial markers (E-cadherin) to mesenchymal cells acquiring the mesenchymal markers (N-cadherin and Vimentin). This transition can promote OSCC metastasis by enhancing migration and invasion [41,68]. A recent study unveiled that the enhanced JAK2/STAT3 pathway caused the EMT of OSCC cells, and EMT could be inhibited by the JAK2 inhibitor AG490 [69]. The activated JAK2/STAT3 signaling pathway decreased E-cadherin expression and increased N-cadherin and E-box binding zinc finger protein 2 (ZEB2) expression in OSCC cells, which induced OSCC growth and metastasis [70]. Furthermore, the role of STAT3 in promoting OSCC cell migration and invasion was also linked to the upregulated expression of matrix metalloproteinase 9 (MMP-9), MMP-7, E-box binding zinc finger protein 1 (ZEB1) [66,71,72]. For example, interleukin-22 (IL-22) and IL-6 were reported to promote the migration and invasion of OSCC cells by activating the JAK/STAT3/MMP-9 signaling pathway [66,72,73]. Moreover, the cytokine-inducible Src homology 2-containing protein (CISH) and SOCS1 increased metastasis of OSCC through promoting the activation of STAT3 [74,75].

Consistent with the above findings, the inhibition of the JAK2/STAT3 may significantly suppress p-STAT3-induced migration and invasion of OSCC cells. Terminal differentiation-induced non-coding RNA (TINCR) suppressed migration of OSCC by highly reducing the expression of JAK2, p-JAK2, STAT3, and p-STAT3 [76]. Aldehyde dehydrogenase 3A1 (ALDH3A1) and miR-144-3p acted as OSCC metastasis suppressors and inhibited EMT via downregulating the STAT3 signaling pathway in OSCC [77,78].

### 3.3. Role of STAT3 in Angiogenesis of OSCC

The pro-angiogenic role of STAT3 has been partially attributed to the upregulation of IL-8, MMP-9, vascular endothelial growth factor (VEGF), angiopoietin 2 (Angpt2), and hypoxia-inducible factor 1-alpha (HIF-1α) via p-STAT3 transactivation [79,80,81]. Recent studies demonstrated that CCL4 increased VEGF-C and Angpt2 production via activating the JAK2/STAT3 signaling pathways in OSCC cells, which was implicated in cell lymph angiogenesis and angiogenesis in OSCC [80,81]. In addition, PA28γ stimulated OSCC tumor angiogenesis in an IL-6/CCL2/STAT3 axis-dependent manner [82]. Conversely, humanized anti-Interleukin-6 receptor antibody Tocilizumab can bind to the IL-6-binding site of human IL-6R and competitively inhibits IL-6 signaling. A drastic reduction in STAT3 phosphorylation induced by Tocilizumab downregulated the expression of VEGF, IL-8, and MMP-9, which decreased microvessel density and vessel diameter in the OSCC xenograft model [79]. Diosmin also suppressed the abnormal expression of VEGF and HIF-1a to obstruct angiogenesis of buccal pouch carcinogenesis through preventing phosphorylation of JAK1/STAT3 [83].

### 3.4. Role of STAT3 in Chemoresistance of OSCC

It has been observed that blocking STAT3 signaling enhances the anticancer activity of chemotherapies in OSCC, thus endorsing a critical role of STAT3 in regulating the chemosensitivity of OSCC [84]. Overexpression and constitutive activation of the STAT3 signaling pathway have been shown to confer chemoresistance on OSCC cells. For example, cancer stem cell-derived extracellular vesicles (CSC-EVs) promoted chemoresistance, stemness, and the metastatic potential of oral cancer cells by enhancing PI3K/mTOR/STAT3 signaling [85]. To the best of our knowledge, several recent studies have revealed the mechanisms underlying STAT3-mediated chemoresistance in OSCC cell lines. Activation of STAT3 and AKT-mediated GSK3β inactivation upregulated Mcl-1 expression, which enhanced TPF resistance in OSCC [86]. Cisplatin treatment upregulated the programmed death ligand 2 (PD-L2) and drug efflux transporter ABCG2 expression in OSCC cell lines via STAT1/3 activation [87]. Indeed, the chemoresistance of cancer cells has been reported to depend on ABC transporter activity.

### 3.5. Role of STAT3 in Immune Suppression

Previous work suggested that STAT3 was a powerful regulator of tumor immunosuppression, and activation of the JAK2/STAT3/PD-L1 signaling axis played a crucial role in the immune escape of osteosarcoma and cervical cancer [88,89]. In OSCC cells, STAT3 also regulated the expression of PD-L1 [90]. Protein kinase D3 (PKD3) was a key kinase mediating the activation of STAT3. Elevated PKD3 expression promoted PD-L1 expression via activating STAT3 in OSCC [91]. STAT3-mediated PD-L1 upregulation affected the proliferation and functional characteristics of T cells [92]. STAT3 was also activated in tumor-associated immune cells to induce the expression of immune suppression related genes and contribute to immunosuppression in OSCC tumor microenvironment (TME) [93,94].

### 3.6. Role of STAT3 in OSCC Stem Cell Phenotypes

Recently, reports have shown that the JAK2/STAT3 signaling pathway also plays a critical role in the EMT and stemness of OSCC. For example, C-C chemokine receptor 7 (CCR7) and its ligand chemokine ligand 21 (CCL21) were reported to be abnormally abundant in OSCC tissues, and CCR7 expression was correlated with EMT and the stemness of OSCC. The treatment with JAK2 inhibitor AG490 could reduce the promotive effects of CCL21 on OSCC cells colony formation and sphere formation [21]. Moreover, blocking the activation of the Jak/Stat3 pathway significantly suppressed the colony forming, invasion, migration, microsphere forming, and xenograft forming abilities of OSCC cells [95]. Overall, these results demonstrated that the JAK/STAT3 signaling pathway may contribute to the stemness of OSCC cells.

### 3.7. Role of STAT3 in Autophagy of OSCC Cells

STAT3 phosphorylation status is capable of influencing autophagy in OSCC cells. Compound 59, an AMPK activator, inhibited JAK/STAT3 signaling, arrested cells in the G1 phase and promoted autophagy. These findings supported the potential of compound 59 for the treatment of OSCC patients through the suppression of STAT3 pathway [96]. Icaritin significantly inhibited the level of p-STAT3 in a dose- and time-dependent manners, and further suppressed proliferation, promoted apoptosis and autophagy [97]. However, more research is needed to understand the mechanism by which STAT3 affects OSCC autophagy.

### 3.8. Role of STAT3 in Radiosensitivity of OSCC Cells

Recently, increasingly more studies have shown STAT3 to contribute to radioresistance. Matsuoka et al. found that the IL-6/STAT3 pathway was relevant in resistance to radiation [98], and overexpression of STAT3 reduced the radiosensitivity of OSCC cells [99]. Yu et al. showed that the expression of Pre-B-cell leukaemia homeobox 1 (PBX1) was abnormally high in OSCC, and knockdown of PBX1 substantially enhanced sensitivity to radiation in OSCC cells by inhibiting STAT3 expression [100]. These studies suggested that STAT3 was not only involved in tumorigenesis and tumor development, but also lead to radioresistance. STAT3 is emerging as a promising target for radio-sensitization of cancer radiotherapy. The mechanism underlying STAT3-mediated radioresistance was related to the suppression of apoptosis and DNA damage induced by STAT3 signaling after radiotherapy [101]. STAT3 enhanced the transcription of apoptosis regulator Mcl-1 and cell cycle regulator cyclin D2 (CCND2) to decrease the sensitivity of cancer cells to radiation, according to more research on the glioma [102], as well as colorectal cancer [103].

### 3.9. Role of STAT3 in Immune Cells within the OSCC TME

TME is a highly complex and heterogenous ecosystem consisting of tumor infiltrating cancer cells, immune cells, and other cells. STAT3 is not only activated in cancer cells, but also becomes hyperactivated in immune cells within the OSCC TME. Some studies have suggested that activation of the JAK2/STAT signaling pathway impaired the antitumor activity of immune cells. For example, radiation treatment of head and neck cancer induced STAT3 signaling to enhance the abundance and function of regulatory T (Treg) cells and resistance to radiation therapy. STAT3 inhibition was beneficial in patients receiving radiation therapy [104]. IL-6-induced p-STAT3 increased PD-L1 and IL-10 expression in myeloid-derived suppressor cells (MDSCs), which impeded proliferation and activation of T cells and promoted Th17 cells differentiation in OSCC [94], and p-STAT3 inhibition (JSI-124) could alleviate immune suppression [42]. In addition, activation of STAT3 helped the immunosuppressive polarization of tumor-associated macrophages, which contributed to the tumor development [105]. On the contrary, overproduction of SOCS3 in dendritic cells can inhibit JAK2/STAT3 activation, as well as the differentiation and immune activity of dendritic cells, leading to their impaired antitumor effects [106]. The inconsistent role of STAT3 signaling in immune cells may be related to the difference in mechanism of action.

## 4. Targeting STAT3 for OSCC Prevention and Therapy

Abundant evidence has suggested that STAT3 may be a promising molecular target for OSCC treatment [107,108,109]. Targeting a druggable site on STAT3 or inhibiting the function of other proteins involved in the STAT3-dependent signaling cascade can directly or indirectly limit STAT3 signaling [28]. In this section, we discuss the current STAT3-targeting strategies (as shown in Figure 2) for treating and preventing OSCC, as well as the challenges in developing more specific and effective STAT3 inhibitors.

### 4.1. Target Upstream Regulators of STAT3

Various STAT3 inhibitors have been developed and shown some efficacy in OSCC in vitro and in vivo, which are showed in Table 1. A majority of STAT3 inhibitors have been identified to target the upstream regulators of STAT3 signaling. STAT3 is widely expressed and is transiently activated in response to EGF and IL-6. Therefore, small molecules and natural products that were able to inhibit IL-6 secretion and production, e.g., the highly pure neem leaf extract SCNE [119], magnolol [120], honokiol [107,108], and 2-O-Methylmagnolol [121], showed significant inhibitory effects on STAT3 signaling in OSCC. However, most of them also inhibited other signaling pathways in cancer cells and indicated a low level of specificity in targeting the STAT3 signaling pathway. As STAT3 is downstream to several cytokine and growth factor receptors and their associated JAKs, inhibiting JAKs by small molecular inhibitors represents a promising therapeutic option in OSCC. JAK2 inhibitors, including Licochalcone C [122] licochalcone D [123] and licochalcone H [124] were found to inhibit OSCC cell viability and induce apoptosis through tightly interacting with ATP-binding site of JAK2 and inhibiting the JAK2/STAT3 signaling pathway. In addition, all-trans retinoic acid (ATRA) showed great potential in cancer treatment through inhibition of p-STAT3 and p-JAK2 [125]. In recent years, Roxyl-ZR [126] and Alkannin [127] presented antitumor activities by hindering JAK1/STAT3 pathways in OSCC pathogenesis.

Targeting the intrinsic STAT3 inhibitors, such as SOCS and PTPs, has been considered as a potential strategy for repressing STAT3 signaling pathway. For example, Bovine lactoferrin attenuated the growth of OSCC through increasing SOCS3 activation and then enhanced SOCS3-mediated STAT3 dephosphorylation and inactivation [132]. A microtubule inhibitor MPT0B098 suppressed the JAK2/STAT3 signaling pathway through modulation of SOCS3 stability in OSCC, which lead to sensitization of OSCC cells to MPT0B098 cytotoxicity [84]. Moreover, inactivated eIF5A2 induced by N1-guanyl-1,7-diaminoheptane (GC7) increased cisplatin chemosensitivity in OSCC cells via inhibition of the STAT3 signaling pathway [131]. Metformin also suppressed the invasion and migration of OSCC through the inhibition of PKM2/STAT3 [136,137]. Although many small molecule agents targeting the upstream regulators exhibited potential antitumor effects through the inhibition of STAT3, other off-target pathways were also activated. Therefore, more careful and thorough pre-clinical investigations must be implemented to prevent potential harmful effects.

### 4.2. Directly Bind to STAT3 and Inhibit Its Activation

STAT3 inhibitors have been developed to interact with the STAT3 domains, with a major focus on the SH2 domain and the DBD. For instance, N-4-hydroxyphenylretinamide (4-HPR) was highly bound at STAT3’s dimerization site and c-Abl and c-Src ATP-binding kinase sites to suppress cancer-promoting pathways including STAT3 phosphorylation, STAT3-DNA binding, and production of the trans-signaling enabling sIL-6R [138]. The STAT3 inhibitor compound (Stattic) was an SH2-domain inhibitor discovered by a high-throughput chemical library screen [109], and Stattic significantly attenuated EZH2 expression and local tumor invasion and outgrowth via targeting STAT3 [139]. However, STAT3 SH2-domain inhibitors moved slowly into clinical medicine due to high homology of the SH2 domain between STAT3 and other family members as well as high concentrations required for disruption of protein–protein interactions, which increased off-target toxicities [146].

### 4.3. Inhibit STAT3 Phosphorylation

Other STAT3 inhibitors were found to inhibit STAT3 phosphorylation, whereas they have not been investigated for the binding ability with STAT3 and ability to regulate upstream regulators of STAT3. For example, Niclosamide inhibited the migration and invasion in OSCC cells through the downregulation of p-STAT3 at Tyr705 [110]. Nitidine chloride acted as an apoptosis inducer in OSCC cells via inhibiting the phosphorylation of STAT3 and transcription of target genes [141,142]. Aspirin and anoctamin1 also suppressed the phosphorylation of STAT3 [140,145]. However, the mechanism of phosphorylation suppression needs to be further studied.

### 4.4. Nuclear-Targeted siRNA Delivery for STAT3 Gene Silencing

The STAT3 inhibitors we introduced above face many problems in their application. For example, how to target cancer cells or immune cells to improve the efficacy; the low cell permeability also prevents the inhibitors from being fully utilized; and the toxic side effects caused by off-target effects need to be addressed. Specificity and selectivity of these inhibitors of STAT3 have been questioned. Currently, a growing number of preclinical and clinical studies showed that antisense oligonucleotide (ASO)-induced STAT3 silence became a promising approach for lymphoma and lung cancer therapy. However, ASOs targeting STAT3 may cause thrombocytopenia, and toxicity limitations hindered the development of drugs [111,112]. In addition, STAT family members share a high degree of structural similarity with each other, which may induce the silence of other family members [147]. Moreover, transcription factors in different cell types usually regulate different gene networks and cellular functions, so how to target the specific cells is also crucial for the development of ideal therapeutic agents. The latest study suggested that ASO targeting STAT6 encapsulated in engineered exosome (exoASO-STAT6) was able to selectively silence STAT6 expression in macrophages [148]. This result may provide an inspiration to develop novel siRNA and antisense technology, which can specifically target STAT3 in tumor cells.

### 4.5. Regulate Downstream Targets of STAT3

As mentioned above, inhibitors targeting STAT3 or its upstream regulators may cause off-target effects, which are connected to the following factors: cell membranal or intracellular signals may deliver in a network, not in a single track; structural conservation of STAT family members; and diversity of STAT3 target genes. These underscore the need to develop inhibitors to regulate downstream targets of STAT3. Precise obstruction of STAT3 downstream signals could help prevent the unintended consequences of general STAT3 inhibition. For example, STAT3 has been reported to act as a transcription factor for Nicotinamide N-methyltransferase (NNMT), and upregulated NNMT in OSCC can contribute to proliferation and invasiveness [149,150]. Newly discovered NNMT inhibitors may be further proposed for OSCC treatment [113,114,115]. In addition, STAT3 was involved in the positive regulation of Sox4, Mcl-1, and so on. Genetic (siRNA) or pharmacological (Triptolide) inhibition of those targets suppressed OSCC growth in vivo [86,118]. Above all, the targeted molecules usually participate in complex signaling pathways. Although some STAT3 inhibitors have entered clinical trials in human OSCC or other malignancy (as shown in Table 2), future study should fully understand the role of STAT3 signaling in OSCC progression, which will aid in the discovery of more specific targeted therapeutics.

## 5. Conclusions

Although some progress has been made in the treatment of OSCC, its prognosis remains poor, with a 5-year survival rate of nearly 50% [10]. A comprehensive understanding of the genetic and molecular disorders of OSCC is critical for early diagnosis, appropriate treatment, and improved prognosis of patients with OSCC. The STAT3 oncogene has been reported to be overexpressed and constitutively activated in OSCC, and is associated with the poor survival outcomes [23,58]. STAT3 drives proliferation and anti-apoptosis, migration and invasion, chemoradiotherapy resistance and angiogenesis, as well as immune evasion of OSCC cells, which highlights the enormous therapeutic potential of STAT3 inhibitors.

Interestingly, contrary to its accepted tumor-promoting role, a fraction of research has shown an opposite role of STAT3 in cancer cells. It was observed that STAT3 knockdown promoted the growth of MDA-MB-231 cell-derived xenograft tumors [175]. Another finding also supported the role of STAT3 activation as a marker of favorable outcome in ER-positive/HER2-positive breast cancer patients [176]. These results imply that the oncogenic role of STAT3 in tumors might be context specific. STAT3 may be a negative regulator of certain cancer types, and therapies targeting STAT3 may therefore need to consider the origin of the tumor type.

Many STAT3-targeted therapies have been successfully developed and have shown efficacy in preclinical models of OSCC in vitro and in vivo. However, cancers including OSCC are multifaceted heterogeneous diseases [177]. OSCC has been reported to harbor multiple genetic alterations, which contribute to its initiation and progression. Therefore, STAT3 inhibition combined with other targeted therapies may be more effective for OSCC. Furthermore, many small molecule agents with potential therapeutic effects for OSCC should be further studied with a view to achieving precise drug delivery, avoiding drug side effects, and translating research findings into clinical applications. In summary, it is essential to have a comprehensive understanding of STAT3 and develop the effective approaches to inhibit the STAT3 signaling pathway with high selectivity and specificity for the treatment of OSCC.

## Figures and Tables

**Figure 1 cells-11-03131-f001:**
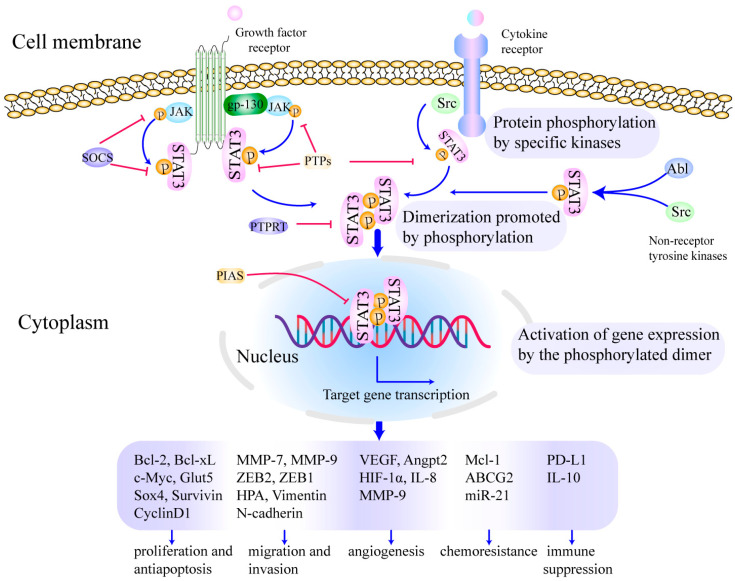
Activation of STAT3 signaling promotes growth, metastasis, chemoresistance, immune suppression and angiogenesis in OSCC. The cytokines or growth factors bind to the corresponding receptors on the cell membrane, which further prompts STAT3 activation and subsequent regulates transcriptional activity. The whole process is divided into three steps as follows: (**1**) protein phosphorylation by specific kinases [19,26,27], (**2**) dimerization promoted by phosphorylation [51], (**3**) activation of gene expression by the phosphorylated dimer [52,53]. Finally, transcription and translation of the target gene regulate cell proliferation and anti-apoptosis, migration and invasion, chemoradiotherapy resistance and angiogenesis, as well as immune suppression in OSCC [22,54,55]. Reproduced/adopted in modified form from [56].

**Figure 2 cells-11-03131-f002:**
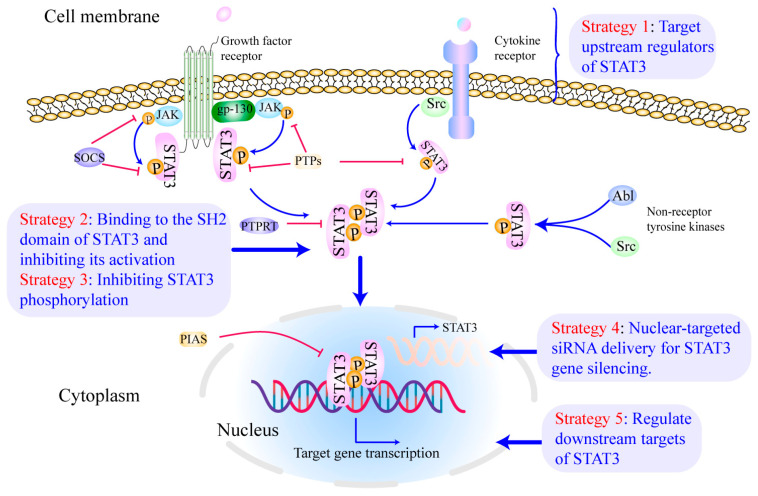
Inhibiting STAT3 signaling at multiple levels for OSCC treatment. Currently, the majority of STAT3 inhibitors have been developed to target different sites in the STAT3 pathway: (**1**) targeting the upstream regulators of STAT3 [110,111], (**2**) binding to the SH2 domain of STAT3 and inhibiting its activation [112], (**3**) inhibiting STAT3 phosphorylation [113], (**4**) nuclear-targeted siRNA delivery for STAT3 gene silencing [114,115], (**5**) regulating downstream targets of STAT3 [116,117,118]. Reproduced/adopted in modified form from [56].

**Table 1 cells-11-03131-t001:** Summary of STAT3 inhibitors and their mechanisms of action for OSCC treatment.

Inhibitors	Mechanisms of Action	In Vitro Activity	In Vivo Activity	References
Target upstream regulators of STAT3				
SCNE	Inhibits IL-6/STAT3 signaling pathway	Inhibits cell proliferation and migration and reduces MMP activity (SCC4, CAL27, HSC3)	Suppresses tumor growth	[119]
MM1	Inhibits IL-6/STAT3 signaling pathway	Suppresses invasion and migration (SAS, OECM1)	Suppresses tumor growth	[121]
Honokiol	Inhibits IL-6/STAT3 signaling pathway	Suppresses cell migration, induces apoptosis, and sensitizes cells to chemotherapy (SAS, OECM1)	Suppresses tumor growth	[107,108]
Magnolol	Inhibits IL-6/STAT3 signaling pathway	Downregulates the self-renewal and metastasis potential of OSCC-CSCs (SAS, GNM)	NR	[120]
Diosmin	Inhibits IL-6/JAK1/STAT3 signaling pathway	NR	Suppresses tumor growth	[83]
Curcumin	Inhibits EGFR/STAT3 signaling pathway	Inhibits proliferation and invasion (SCC25)	NR	[128]
Alkannin	Inhibit JAK1/STAT3 signaling pathway	Restrains cell growth, migration and invasion, and facilitates apoptosis (KB)	Suppresses tumor growth	[127]
Roxyl-ZR	Inhibits JAK1/STAT3 signaling pathway	Inhibits metabolism, clonogenicity, proliferation, migration and invasion (UM1, TSCCA)	Suppresses tumor growth	[126]
All-trans retinoic acid	Inhibits JAK2/STAT3 signaling pathway	Inhibits proliferation and induces Apoptosis (CAL27, DOK)	NR	[125]
Licochalcone H	Inhibits JAK2/STAT3 signaling pathway	Inhibits cell growth and induces apoptosis (HN22, HSC4)	NR	[124]
Licochalcone D	Inhibits JAK2/STAT3 signaling pathway	Inhibits the cell growth and colony formation (HN22, HSC4)	Suppresses tumor growth	[123]
Compound 59	Inhibits JAK2/STAT3 signaling pathway	Induces autophagy and apoptosis (SCC2095, SCC4)	NR	[96]
β-Elemene	Inhibits JAK2/STAT3 signaling pathway	Inhibits proliferation and induces apoptosis (Tca8113)	Suppresses tumor growth	[129]
Icaritin	Inhibits JAK2/STAT3 signaling pathway	Induces autophagy and apoptosis (CAL27, SCC9)	Suppresses tumor growth	[97]
Trichodermin	Inhibits STAT3 signaling pathway	Inhibits proliferation, migration and invasion (Ca922, HSC3)	Suppresses tumor growth	[130]
MPT0B098	Stabilize SOCS3	Inhibits growth and induces apoptosis (OECM1)	NR	[84]
GC7	Inhibits eIF5A2/STAT3 signaling pathway	Sensitizes OSCC cells to cisplatin (CAL27, HN4, HN30, Tca8113)	NR	[131]
Bovine Lactoferrin	Stabilize SOCS3	Inducts apoptosis, and suppresses Proliferation (HSC3)	NR	[132]
Licochalcone C	Inhibits JAK2/STAT3 signaling pathway	Induces apoptosis (HN22, HSC4)	NR	[122]
Betulinic acid	Inhibits STAT3 signaling pathway	Inhibits cell proliferation (KB, SAS)	Suppresses tumor growth	[133,134]
Koetjapic acid	Inhibits STAT3 signaling pathway	Inhibits proliferation, invasion, angiogenesis, and metastasis (SAS)	NR	[134]
Isoorientin	Blocking Wnt/β-catenin/STAT3 axis	Attenuates OSCC cell stemness and EMT potential (SAS, SCC25)	Suppresses tumor growth	[135]
Metformin	Inhibits mTOR/HIF-1α/PKM2/STAT3 pathway	Inhibits proliferation, migration and invasion (CAL27)	Suppresses tumor growth	[136]
	Inhibits STAT3/TWIST pathway	Inhibits invasion and migration (HSC3, HSC6)	NR	[137]
Strategy 2: Directly bind to STAT3 and inhibit its activation				
4-HPR	Binds to STAT3 and inhibits its phosphorylation	Inhibits proliferation (JSCC1, JSCC2, JSCC3)	Suppresses tumor growth	[138]
Stattic	Binds to SH2 domain of STAT3 and inhibits phosphorylation	Inhibits invasion and migration (SCC15, SCC25)	Suppresses tumor metastasis	[109,139]
Strategy 3: Inhibit STAT3 phosphorylation				
Aspirin	Inhibits STAT3 phosphorylation	Induces the cell cycle arrest and apoptosis, and suppresses cell migration and invasion (Tca8113, CAL27)	NR	[140]
Nitidine chloride	Inhibits STAT3 phosphorylation	Induces apoptosis, and suppresses proliferation	Suppresses tumor growth	[141,142]
Niclosamide	Inhibits STAT3 phosphorylation	Suppresses proliferation, migration and invasion (HSC3, HSC4, WSU-HN6, Tca83)	NR	[110]
Bupivacaine	Inhibits STAT3 phosphorylation	Promotes apoptosis (CAL27)	Suppresses tumor growth	[143]
WP1066	Inhibits STAT3 phosphorylation	Suppresses proliferation, migration and invasion (TSCCA, Tca8113)	Suppresses tumor growth	[144]
Anoctamin1	Inhibits STAT3 phosphorylation	Reduces cell proliferation and migration (CAL27)	NR	[145]

NR, not reported.

**Table 2 cells-11-03131-t002:** Summary of STAT3 inhibitors in clinical trials.

Inhibitors	Target	NCT Number	Conditions	Phase	Reference
STAT3 DECOY	STAT3	NCT00696176	Head and neck cancer	Phase I	[151]
Pyrimethamine	STAT3	NCT01066663	Chronic lymphocytic leukemia	Phase II	[152]
Tipifarnib	STAT3	NCT00049114	IIB-IIIC breast cancer	Phase II	[153]
OPB-31121	STAT3	NCT00657176	Solid tumor	Phase I	[154]
		NCT00955812	Solid tumor	Phase I	[155]
Danvatirsen	STAT3	NCT02549651	Relapsed or refractory diffuse large B-cell lymphoma	Phase I	[156]
MSC-1 (AZD0171)	STAT3	NCT03490669	Advanced solid tumors	Phase I	[157]
OPB-51602	STAT3	NCT01184807	Solid malignancies	Phase I	[158]
AZD9150	STAT3	NCT01563302	Lymphoma	Phase I	[112]
WP1066	STAT3	NCT02977780	Recurrent malignant glioma	Phase I	[159]
Erlotinib	EGFR	NCT00779389	Head and neck cancer	Phase I	[160]
Napabucasin	STAT3	NCT01830621	Advanced colorectal cancer	Phase III	[161]
Napabucasin	STAT3	NCT02753127)	Metastatic colorectal cancer	Phase I	[162]
OPB-111077	STAT3	NCT01942083	Advanced hepatocellular carcinoma	Phase I	[163]
		NCT01711034	Advanced cancer	Phase I	[164]
Ruxolitinib	JAK1/2	NCT02041429	HER2-negative metastatic breast cancer	Phase I	[165]
		NCT02145637	Non-small cell lung cancer	Phase I	[166]
		NCT02015208	Chronic lymphocytic leukemia	Phase II	[167]
		NCT02066532	Metastatic HER2 positive breast cancer	Phase I II	[168]
		NCT00674479	Postmyeloproliferative neoplasm acute myeloid leukemia	Phase II	[169]
		NCT01702064	Chronic myeloid leukemia	Phase I	[170]
AZD1480	JAK2	NCT01112397	Solid tumor	Phase I	[171]
CEP-701	JAK2	NCT00494585	Primary or post-polycythemia vera/essential thrombocythemia myelofibrosis	Phase II	[172]
Afatinib	EGFR	NCT02145637	EGFR mutant NSCLC	Phase I	[166]
Nilotinib	EGFR	NCT01168050	KIT-Altered Melanoma	Phase II	[173]
nilotinib	EGFR	NCT01061177	Chronic myeloid leukemia	Phase III	[174]

## Data Availability

Not applicable.

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
