# Peer review of "STAT3 and Its Targeting Inhibitors in Oral Squamous Cell Carcinoma"

_cells, 2022, doi:10.3390/cells11193131_

Round 1

Reviewer 1 Report

In this manuscript, the authors summarized the recent research progress of STAT3 in OSCC, focusing on the oncogenic role of STAT3 in OSCC, and outline the therapeutic strategies and representative small molecule inhibitors targeting STAT3. Overall, the authors provide a fair and unbiased appraisal of the subject. But there are still some points should be addressed before it can be accepted for publication.

1.In the Figure Legends, the references should be indicated to show where the figures come from. 

2.The effect of drugs on different cell lines may vary due to the different genetic background. It is better to show the names of cell lines in the In Vitro Activity in Table 1.

3. More clinical trials can be found the websites such as https://clinicaltrials.gov/. Please include these information in Table 2.

Author Response

Response to Reviewer 1 Comments

Point 1: In the Figure Legends, the references should be indicated to show where the figures come from.  

Response 1: In the Figure Legends, the references have been indicated to show where the figures come from (Figure 1, page 4, line 134; Figure 2, page 7, line 295).

Point 2: The effect of drugs on different cell lines may vary due to the different genetic background. It is better to show the names of cell lines in the In Vitro Activity in Table 1.

Response 2: The names of cell lines have been shown in the In Vitro Activity in Table 1 (pages 8-9).

Point 3: More clinical trials can be found the websites such as https://clinicaltrials.gov/. Please include these information in Table 2.

Response 3: More clinical trials have been included in Table 2 (page 11).

Reviewer 2 Report

the review is well written and important. still, several issues should be addressed. 

Comments:

1.       Line 47 – Provide references to the statement, “Recently, there is growing evidence that signal transducer 44 and activator of transcription 3 (STAT3) is overexpressed and constitutively activated in 45 OSCC and has an important role in several aspects of cancer aggressiveness

2.       Line 70-71 Provide references – “Recent evi- 69 dence has demonstrated that STAT3 plays a critical role in OSCC and STAT3 inhibitors have 70 shown efficacy in inhibiting OSCC tumor growth and metastasis

3.       Line 103; where was it found, what kind of cancer? And what exactly it induces? Provide references

4.       Lines 126 – 129, the authors state that STAT3 is overexpressed in OSCC. However, they are not clarifying its role or its effect on other proteins and signals.

5.       In lines 169 – 183, it would be beneficial to add the exact role of STAT3 in EMT. Since EMT is the major process for metastasis development, it is not clear how STAT3 is involved in EMT.

6.       Line 187 reference?

7.       Line 210 reference

8.       Line 236, what recent studies??? Reference

9.       The part: Role of STAT3 in radiosensitivity of OSCC cells – is not clear. It is not clear how STAT3 is associated with radiosensitivity… the authors should explain this part and ad more studies to support this statement.  

10.   The statement “ The STAT3 oncogene is  overexpressed and constitutively activated in OSCC and is associated with the poor survival outcomes.” Is not supported by the review. The authors should add more studies to support this conclusion.

Author Response

Response to Reviewer 2 Comments

Point 1: Line 47 – Provide references to the statement, “Recently, there is growing evidence that signal transducer and activator of transcription 3 (STAT3) is overexpressed and constitutively activated in OSCC and has an important role in several aspects of cancer aggressiveness”

Response 1: The corresponding references to the above statement (Line 47) have been provided (page 2, line 76).  

Point 2: Line 70-71 Provide references – “Recent evidence has demonstrated that STAT3 plays a critical role in OSCC and STAT3 inhibitors have shown efficacy in inhibiting OSCC tumor growth and metastasis”

Response 2: The corresponding references to the above statement (Line 70-71) have been provided (page 2, line 79).

Point 3: Line 103; where was it found, what kind of cancer? And what exactly it induces? Provide references

Response 3: To show the answers to the above questions, the statement (Line 103) has been revised as follows: STAT3-mediated IP3R3 downregulation in the ER was a major factor in anti-apoptotic effects of breast cancer cell lines[36]. These results suggest that a further non-canonical function played by constitutively phosphorylated STAT3 at its S727 residue promotes the growth of cancer and inhibits apoptosis, but the mechanism remains to be clarified[32, 36, 37] (page 3, lines 113-117).

References:

  1. Gough, D.J., et al., Mitochondrial STAT3 supports Ras-dependent oncogenic transformation. Science, 2009. 324(5935): p. 1713-6.
  2. Avalle, L., et al., STAT3 localizes to the ER, acting as a gatekeeper for ER-mitochondrion Ca(2+) fluxes and apoptotic responses. Cell Death Differ, 2019. 26(5): p. 932-942.
  3. Gough, D.J., et al., STAT3 supports experimental K-RasG12D-induced murine myeloproliferative neoplasms dependent on serine phosphorylation. Blood, 2014. 124(14): p. 2252-61.

Point 4: Lines 126 – 129, the authors state that STAT3 is overexpressed in OSCC. However, they are not clarifying its role or its effect on other proteins and signals.

Response 4: The role of the overexpressed STAT3 has been added in the revised statement, but its effects on other proteins and signals were not investigated in this study. The statement (Lines 126 – 129) has been revised as follows: In addition, endogenous negative regulators such as PTPN4, SOCS5 and SOCS6 were decreased in OSCC cells, which contributed to the upregulation of p-STAT3. The upregulated p-STAT3 promoted the OSCC progression [48, 49]” (page 4, lines 148-151).

References:

  1. Liu, X., Y. Dong, and D. Song, Inhibition of microRNA-15b-5p Attenuates the Progression of Oral Squamous Cell Carcinoma via Modulating the PTPN4/STAT3 Axis. Cancer Manag Res, 2020. 12: p. 10559-10572.
  2. Tan, J., L. Xiang, and G. Xu, LncRNA MEG3 suppresses migration and promotes apoptosis by sponging miR-548d-3p to modulate JAK-STAT pathway in oral squamous cell carcinoma. IUBMB Life, 2019. 71(7): p. 882-890.

Point 5: In lines 169 – 183, it would be beneficial to add the exact role of STAT3 in EMT. Since EMT is the major process for metastasis development, it is not clear how STAT3 is involved in EMT.

Response 5: The exact role of STAT3 in EMT has been added in the section of “Role of STAT3 in OSCC cell migration and invasion” as follws: Epithelial-mesenchymal transition (EMT) is an important biological mechanism underlying the metastasis of primary Tumors[68]. During the EMT, the characteristic of epithelial cells will convert from epithelial cells highly expressing epithelial markers (E-cadherin) to mesenchymal cells acquiring the mesenchymal markers (N-cadherin and Vimentin). This transition can promote OSCC metastasis by enhancing migration and invasion[42, 69]. A recent study unveiled that the enhanced JAK2/STAT3 pathway caused EMT of OSCC cells, and EMT could be inhibited by the JAK2 inhibitor AG490[70]. The activated JAK2/STAT3 signaling pathway decreased E-cadherin expression and increased N-cadherin and E-box binding zinc finger protein 2 (ZEB2) expression in OSCC cells, which induced OSCC growth and metastasis[71]. Furthermore, the role of STAT3 in promoting OSCC cells migration and invasion was also linked to the upregulated expression of matrix metalloproteinase 9 (MMP-9), MMP-7, E-box binding zinc finger protein 1 (ZEB1)[67, 72, 73]. For example, interleukin-22 (IL-22) and IL-6 were reported to promote the migration and invasion of OSCC cells by activating the JAK/STAT3/MMP-9 signaling pathway[67, 73, 74]. Besides, the cytokine-inducible Src homology 2-containing protein (CISH) and SOCS1 increased metastasis of OSCC through promoting activation of STAT3[75, 76]. (pages 4-5, lines 173-190).

Point 6: Line 187 reference?

Response 6: The corresponding references (Line 187) have been provided (page 5, line 186).

References:

  1. Liu, Z., et al., STAT3-induced upregulation of long noncoding RNA HNF1A-AS1 promotes the progression of oral squamous cell carcinoma via activating Notch signaling pathway. Cancer Biol Ther, 2019. 20(4): p. 444-453.
  2. Hsieh, Y.P., et al., Epigenetic Deregulation of Protein Tyrosine Kinase 6 Promotes Carcinogenesis of Oral Squamous Cell Carcinoma. Int J Mol Sci, 2022. 23(9).
  3. Komine-Aizawa, S., et al., Interleukin-22 promotes the migration and invasion of oral squamous cell carcinoma cells. Immunol Med, 2020. 43(3): p. 121-129.

Point 7: Line 210 reference

Response 7: The corresponding references (Line 210) have been provided (page5, line 200). The statement has been revised as follows: The pro-angiogenic role of STAT3 has been partially attributed to the upregulation of IL-8, MMP-9, vascular endothelial growth factor (VEGF), angiopoietin 2 (Angpt2) and hypoxia-inducible factor 1-alpha (HIF-1α) via p-STAT3 transactivation[80-82] (page 5, lines 198-200).

References:

  1. Shinriki, S., et al., Humanized anti-interleukin-6 receptor antibody suppresses tumor angiogenesis and in vivo growth of human oral squamous cell carcinoma. Clin Cancer Res, 2009. 15(17): p. 5426-34.
  2. Lu, C.C., et al., The Chemokine CCL4 Stimulates Angiopoietin-2 Expression and Angiogenesis via the MEK/ERK/STAT3 Pathway in Oral Squamous Cell Carcinoma. Biomedicines, 2022. 10(7).
  3. Lien, M.Y., et al., Chemokine CCL4 Induces Vascular Endothelial Growth Factor C Expression and Lymphangiogenesis by miR-195-3p in Oral Squamous Cell Carcinoma. Front Immunol, 2018. 9: p. 412.

Point 8: Line 236, what recent studies??? Reference

Response 8: The statement (Line 236) has been revised as follows: Previous work suggested that STAT3 was a powerful regulator of tumor immunosuppression, and activation of JAK2/STAT3/PD-L1 signaling axis played a crucial role in immune escape of osteosarcoma and cervical cancer[89, 90] (page 5, lines 227-229). The corresponding references have been provided (page 5, line 229).

References:

  1. Jing, D., et al., Quercetin encapsulated in folic acid-modified liposomes is therapeutic against osteosarcoma by non-covalent binding to the JH2 domain of JAK2 Via the JAK2-STAT3-PDL1. Pharmacol Res, 2022. 182: p. 106287.
  2. Zhou, C., et al., Cancer-secreted exosomal miR-1468-5p promotes tumor immune escape via the immunosuppressive reprogramming of lymphatic vessels. Mol Ther, 2021. 29(4): p. 1512-1528.

Point 9: The part: Role of STAT3 in radiosensitivity of OSCC cells – is not clear. It is not clear how STAT3 is associated with radiosensitivity… the authors should explain this part and ad more studies to support this statement.

Response 9: The section of “Role of STAT3 in radiosensitivity of OSCC cells” has been revised to further illustrate the relationship between STAT3 and radiosensitivity as follows: Recently, more and more studies showed that STAT3 contributed to radioresistance. Matsuoka et al. found that IL-6/STAT3 pathway was relevant to resistance to radiation[99], and overexpression of STAT3 reduced the radiosensitivity of OSCC cells[100]. Yu et al. showed that the expression of Pre-B-cell leukemia homeobox 1 (PBX1) was abnormally high in OSCC, and knockdown of PBX1 substantially enhanced sensitivity to radiation in OSCC cells by inhibiting STAT3 expression[101]. These studies suggested that STAT3 was not only involved in tumorigenesis and tumor development but also lead to radioresistance. STAT3 is emerging as a promising target for radio-sensitization of cancer radiotherapy. The mechanism underlying STAT3-mediated radioresistance was related to the suppression of apoptosis and DNA damage induced by STAT3 signaling after radiotherapy[102]. STAT3 enhanced the transcription of apopto-sis regulator Mcl-1 and cell cycle regulator cyclin D2 (CCND2) to decrease the sensitivity of cancer cells to radiation, according to more research on the glioma[103], as well as colorectal cancer[104]. (page 6, lines 250-270).

Point 10: The statement “ The STAT3 oncogene is overexpressed and constitutively activated in OSCC and is associated with the poor survival outcomes.” Is not supported by the review. The authors should add more studies to support this conclusion.

Response 10: The above statement is from references. The corresponding references have been provided (page 12, line 403). The statement has been revised as follows: It was reported that the STAT3 oncogene was overexpressed and constitutively acti-vated in OSCC, and associated with the poor survival outcomes[24, 59] (page 12, lines 403-405).

References:

  1. Macha, M.A., et al., Prognostic significance of nuclear pSTAT3 in oral cancer. Head Neck, 2011. 33(4): p. 482-9.
  2. Wei, L.Y., et al., Effects of Interleukin-6 on STAT3-regulated signaling in oral cancer and as a prognosticator of patient survival. Oral Oncol, 2022. 124: p. 105665.

Reviewer 3 Report

The manuscript “STAT3 and its targeting inhibitors in oral squamous cell carcinoma” is a review article regarding the actual state-of-art on the importance of STAT3-dependent functions in OSCC and the developing of STAT3 inhibitors for the treatment of OSCC. There are some important concerns that authors should address before the manuscript could be considered for publication:

1.       Sometimes there is an awkward construction of sentences. A language revision is highly recommended.

2.       “in vitro” and “in vivo” must be written in italics throughout the text.

3.       In the paragraph “background” some parts are written with different font styles. Please reconcile.

4.       The background of the review is scarce and does not provide enough information regarding the OSCC (incidence, classification, aggressive subtypes, etc..). Thus, authors should expand this section accordingly.

5.       Table 1 and table 2 are somehow compressed and difficult to be read. Columns and lines should be more spaced.

6.       The review includes a paragraph “Target upstream regulators of STAT3” and other paragraphs in which direct inhibition of STAT3 is discussed. However, since authors discuss the importance of STAT3 signaling in OSCC progression, a paragraph underlining the actual research in developing inhibitors downstream STAT3 activation is required. This is of particular importance, since a fine blockade of downstream targets would prevent detrimental effects of a general STAT3 inhibition, which would affect also the functions of normal cells. For instance, STAT3 has been reported to act as a transcription factor for the enzyme Nicotinamide N-methyltransferase (NNMT) (PMID: 17922140) which has been demonstrated to be upregulated in OSCC where contributes to proliferation and aggressiveness (PMID: 29882109). Furthermore, there is an intensive research focused on developing NNMT inhibitors (PMID: 34572571; PMID: 34704059; PMID: 34424711) which may be proposed also for the OSCC management. Other genes upregulated by Stat3 in OSCC are Sox4 (PMID: 35513871), Mcl-1 (PMID: 30395230).

Author Response

Response to Reviewer 3 Comments

Point 1: Sometimes there is an awkward construction of sentences. A language revision is highly recommended.

Response 1: A language revision has been made to avoid the awkward construction of sentences.

Point 2: “in vitro” and “in vivo” must be written in italics throughout the text.

Response 2: “in vitro” and “in vivo” have been written in italics throughout the text.

Point 3: In the paragraph “background” some parts are written with different font styles. Please reconcile.

Response 3: Different font styles have been reconciled according to the requirement (pages 1-2).

Point 4: The background of the review is scarce and does not provide enough information regarding the OSCC (incidence, classification, aggressive subtypes, etc..). Thus, authors should expand this section accordingly.

Response 4: The information regarding the OSCC (incidence, classification, aggressive subtypes) has been provided, and the section of background has been expanded. This section has been revised as follows: Oral squamous cell carcinoma (OSCC) represents the most frequent form of head and neck squamous cell carcinoma that is the sixth most common group of cancers worldwide [1-3]. OSCC originates from the tongue, palate, floor of mouth, alveolar ridge, buccal mucosa, and other areas of oral cavity, and accounts for about 90% of oral ma-lignancies[4, 5]. In 2020, about 377,713 new patients were diagnosed with the lip and oral cavity cancers, and its number of deaths was around 177,757 worldwide. Most cases were discovered in Asia[6]. The risk factors for OSCC include smoking, excessive alcohol consumption, and betel nut chewing, exposure to carcinogens, immunodeficiency, irradiation, nutrition, genetic susceptibility, as well as viral infections including human papillomavirus and herpes simplex virus[7]. The main biologic activity of OSCC is classified as highly, moderately, or poorly differentiated along with increased aggres-siveness[8]. Histologically, OSCC exhibits grades ranging from well-differentiated keratinizing carcinoma to undifferentiated nonkeratinizing carcinoma, which is more apt to spread[9, 10]. Patients with OSCC are asymptomatic in the early stages, and most patients are diagnosed when OSCC further progresses, resulting in the lower survival rate[11]. Tumor infiltration, lymph node metastasis, and high rates of local recurrence are the main factors leading to death in patients with OSCC[12]. Current treatment options for OSCC include surgery, chemotherapy, radiotherapy, or a combination of therapies, depending on factors such as the extent of the disease and the patient's comorbidities[13]. However, the adverse effects of the treatment still exist. For example, salivary gland hypofunction is a common and permanent adverse effect of radiotherapy to the head and neck[14]. The common complications after selective neck dissection are spinal accessory nerve damage and shoulder dysfunction[15]. Systemic administration of chemotherapeutic drugs emphasizes the need to avoid the systemic undesired side effects. Targeted therapy for OSCC, which consists of immunotherapy, gene therapy and bionic technology, has shown some promise in preliminary clinical studies, but further investigation is needed[16]. (pages 1-2, lines 27-52).

Point 5: Table 1 and table 2 are somehow compressed and difficult to be read. Columns and lines should be more spaced.

Response 5: Table 1 and table 2 have been revised according to the requirement (pages 8-9 and page 11).

Point 6: The review includes a paragraph “Target upstream regulators of STAT3” and other paragraphs in which direct inhibition of STAT3 is discussed. However, since authors discuss the importance of STAT3 signaling in OSCC progression, a paragraph underlining the actual research in developing inhibitors downstream STAT3 activation is required. This is of particular importance, since a fine blockade of downstream targets would prevent detrimental effects of a general STAT3 inhibition, which would affect also the functions of normal cells. For instance, STAT3 has been reported to act as a transcription factor for the enzyme Nicotinamide N-methyltransferase (NNMT) (PMID: 17922140) which has been demonstrated to be upregulated in OSCC where contributes to proliferation and aggressiveness (PMID: 29882109). Furthermore, there is an intensive research focused on developing NNMT inhibitors (PMID: 34572571; PMID: 34704059; PMID: 34424711) which may be proposed also for the OSCC management. Other genes upregulated by Stat3 in OSCC are Sox4 (PMID: 35513871), Mcl-1 (PMID: 30395230).

Response 6: A paragraph underlining the actual research in developing inhibitors applied to downstream targets of STAT3 has been added as follow: As mentioned above, inhibitors targeting STAT3 or its upstream may cause off-target effects, which are connected to the following factors: cell membranal or intracellular signals may deliver in a network, not in a single track; structural conservation of STAT family members; diversity of STAT3 target genes. These underscore the need to develop inhibitors to regulate downstream targets of STAT3. Precise obstruction of STAT3 downstream signals could help prevent the unintended consequences of general STAT3 inhibition. For example, STAT3 has been reported to act as a transcription factor for Nicotinamide N-methyltransferase (NNMT), and upregulated NNMT in OSCC can contribute to proliferation and invasiveness[150, 151]. Newly discovered NNMT inhibitors may be further proposed for OSCC treatment[114-116]. In addition, STAT3 was involved in the positive regulation of Sox4, Mcl-1 and so on. Genetic (siR-NA) or pharmacological (Triptolide) inhibition of those targets suppressed OSCC growth in vivo[87, 119]. Above all, since the targeted molecules usually participate in complex signaling pathways. Future studies should fully understand the role of STAT3 signaling in OSCC progression, which will aid in the discovery of more specific targeted therapeutics. (pages 11-12, lines 380-395).

References:

  1. Maji, S., et al., STAT3- and GSK3β-mediated Mcl-1 regulation modulates TPF resistance in oral squamous cell carcinoma. Carcinogenesis, 2019. 40(1): p. 173-183. (PMID: 30395230)
  2. van Haren, M.J., et al., Macrocyclic peptides as allosteric inhibitors of nicotinamide N-methyltransferase (NNMT). RSC Chem Biol, 2021. 2(5): p. 1546-1555. (PMID: 34704059)
  3. Gao, Y., et al., Potent Inhibition of Nicotinamide N-Methyltransferase by Alkene-Linked Bisubstrate Mimics Bearing Electron Deficient Aromatics. J Med Chem, 2021. 64(17): p. 12938-12963. (PMID: 34424711)
  4. van Haren, M.J., et al., Esterase-Sensitive Prodrugs of a Potent Bisubstrate Inhibitor of Nicotinamide N-Methyltransferase (NNMT) Display Cellular Activity. Biomolecules, 2021. 11(9). (PMID: 34572571)
  5. Xiao, L., et al., Interleukin-6 mediated inflammasome activation promotes oral squamous cell carcinoma progression via JAK2/STAT3/Sox4/NLRP3 signaling pathway. J Exp Clin Cancer Res, 2022. 41(1): p. 166. (PMID: 35513871)
  6. Tomida, M., et al., Stat3 up-regulates expression of nicotinamide N-methyltransferase in human cancer cells. J Cancer Res Clin Oncol, 2008. 134(5): p. 551-9. (PMID: 17922140)
  7. Seta, R., et al., Overexpression of nicotinamide N-methyltransferase in HSC-2 OSCC cell line: effect on apoptosis and cell proliferation. Clin Oral Investig, 2019. 23(2): p. 829-838. (PMID: 29882109)

Round 2

Reviewer 2 Report

the paper was corrected as advised 

Good work 

Reviewer 3 Report

The manuscript is improved and can be published.